# Molecular Dynamics Analysis of $A\beta40$ and $A\beta42$ Peptides and the System $A\beta40 + A\beta42$

## Abstract

Molecular dynamics simulations (MD) of the $A\beta40$ and $A\beta42$ peptides, and mixed $A\beta40 + A\beta42$ system, were performed using the GROMACS package in an aqueous medium containing Na$^+$ and Cl$^-$ ions at physiological pH (7.4). The mixed system was built by placing both peptides in the same water box to probe their intermolecular interactions and conformational behavior in solution. Analysis of RMSD, RMSF,and radius of gyration ($R_g$) indicates that the $A\beta40$ and $A\beta42$ reach conformational equilibrium in aqueous solution and remain structurally stable throughout the simulation. $A\beta42$ presented a more compact conformation, while the mixed $A\beta40 + A\beta42$ system showed higher fluctuations and higher values for RMSD, RMSF and $R_g$. SASA and hydrogen bonding revealed relevant differences between the systems, indicating progressive expansion for $A\beta40$ and gradual compaction for $A\beta42$. The mixed system showed higher structural instability and solvation patterns consistent with combined monomer behavior. The results suggest that the coexistence of $A\beta40$ and $A\beta42$ amplifies conformational instability, which may favor the initial oligomerization associated with neurotoxicity in Alzheimer's Disease.

## Meaningfulness Statement

In this work, we consider a meaningful representation of life to be one that preserves mechanistic links between molecular structure, dynamics, and emergent biological function, so that changes in the representation correspond to physically plausible biochemical events rather than abstract numerical variation. By simulating $A\beta40$, $A\beta42$, and the mixed $A\beta40 + A\beta42$ system at atomistic resolution, our study encodes each peptide not as a static sequence, but as an evolving ensemble of conformations, hydration patterns, and interaction networks that are directly tied to experimentally observed aggregation propensities and neurotoxicity in Alzheimer's Disease. In this sense, RMSD, RMSF, radius of gyration, SASA, and hydrogen-bond networks become meaningful coordinates of "life-like" behavior: they describe how small sequence differences translate into distinct pathways toward, or away from, pathogenic oligomerization. Our work contributes to this direction by showing how molecular dynamics can generate interpretable, time-resolved representations that connect microscopic motions to macroscopic disease mechanisms, offering a physically grounded substrate on which future AI models can reason about pathological versus protective protein states.

## 1 Introduction

The ratio of $A\beta40$ and $A\beta42$ peptides in the brain is an important physicochemical determinant for the onset of Alzheimer's Disease (AD) (Qiu et al., 2015). These peptides are natural byproducts of neuronal metabolism, resulting from the continuous cleavage of the amyloid precursor protein (APP), and differ by only two amino acids that $A\beta42$ possesses in addition to $A\beta40$ (Chen et al., 2017). However, this small structural difference leads to significant changes in their physicochemical properties (Sulatskaya et al., 2022).

$A\beta42$ is the primary molecular agent triggering AD. $A\beta42$ does not exist in isolation, but as a highly aggregable minority fraction within a matrix dominated by $A\beta40$ (Li et al., 2023). Relative increases in $A\beta42$ shift the system toward a state favoring the formation of stable $A\beta42$ oligomeric nuclei,

which then grow autocatalytically and drive downstream pathological events, including inflammation and synaptic dysfunction, leading to AD (Selkoe & Hardy, 2016).

In addition to the relative $A\beta42/A\beta40$ composition, the chemical environment is another determinant for the onset of AD (Nguyen & et al., 2021). Since $A\beta$ aggregation is a process ruled by electrostatic, hydrophobic, and solvation interactions, factors such as pH, ionic strength, the presence of metals, ligands, and interfaces can jointly modulate the peptide's effective solubility, its conformational flexibility, and the relative stability of pre-aggregated states. In this context, molecular dynamics (MD) simulations have proven effective, showing that pH variations alter the size, flexibility, and secondary-structure propensies of $A\beta$ (Tian & Viles, 2022) and modify its metal-ion coordination, thereby tuning self-association(Albrahadi et al., 2025). Furthermore, enhanced-sampling MD studies have demonstrated that environment-dependent conformational transitions critically govern the formation of oligomeric intermediates that drive nucleation (Paul et al., 2021; Wen et al., 2023), while co-aggregation and *cross-seeding* simulations reveal that interactions with other amyloidogenic species and heterogeneous interfaces stabilize $\beta$-sheet–rich assemblies and promote potentially cytotoxic $\beta$-barrel intermediates (Li et al., 2023; Fan et al., 2024; Huang et al., 2024).

The objective of this work was to investigate, using classical molecular dynamics simulations, the structural and conformational behavior of amyloid-beta peptides $A\beta40$ and $A\beta42$, both in isolation and in a mixed system, under physiological conditions. Specifically, we aimed to investigate the dynamic properties of these peptides through the analysis of structural parameters, including the root mean square deviation, residue fluctuations, radius of gyration, solvent accessible surface area, and intramolecular and solvent hydrogen bond networks.

## 2 METHODOLOGY – COMPUTATIONAL METHODS

The NMR structures of $A\beta40$ (PDB code: 1AML) and $A\beta42$ (PDB code: 1Z0Q) were obtained from the RCSB Protein Data Bank (RCSB PDB) (Protein Data Bank, 1996; 2006). Next, the structures were protonated using the APBS-PDB2PQR software at pH 7.4, corresponding to human physiological pH. Subsequently, PyMOL (Schrödinger, 2015) software was used to remove the hydrogens. Furthermore, the mixed $A\beta40 + A\beta42$ system was assembled in PyMOL with an initial intermolecular distance of 10.3 Å, generating a forced initial arrangement favorable for intermolecular interactions (Figure 1).

Figure 1: Preparation steps for the $A\beta40$ and $A\beta42$ peptides and the mixed $A\beta40 + A\beta42$ system. The flowchart illustrates the acquisition of structures from the RCSB PDB, protonation, and assembly in PyMOL for molecular dynamics simulations.

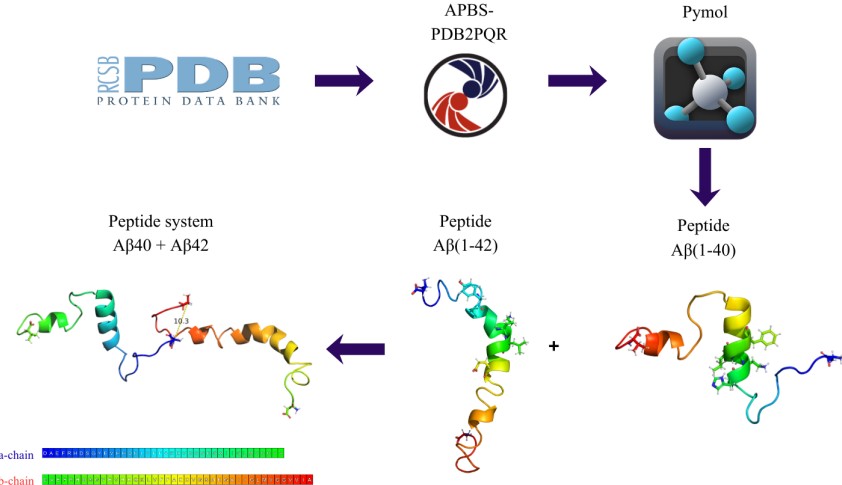

Molecular dynamics (MD) simulations were performed using the GROMACS (Abraham et al., 2015) software package version 2025.4, employing the CHARMM36 (Huang & MacKerell Jr, 2013) force field. The peptides were simulated in an explicit water box using the TIP3P model (Jorgensen

et al., 1983) and ionized with $Na^+$ and $Cl^-$ ions at a concentration of $0.15 \text{ mol L}^{-1}$. Energy minimization was conducted until the maximum force was less than $150.0 \text{ kJ mol}^{-1}$. Equilibration under the NVT ensemble was performed at $300$ K, while NPT equilibration utilized a compressibility of $4.5 \times 10^{-5} \text{ bar}^{-1}$, using the Parrinello–Rahman barostat. Both the minimization and equilibration steps were performed with position restraints applied to the protein. After equilibration, the position restraints were removed, and production molecular dynamics was conducted for $100$ ns (Figure 2).

Figure 2: General steps of the molecular dynamics simulation performed in GROMACS. Flowchart showing the centering of the protein in the box (1.0 nm from the edges), solvation, addition of ions, equilibration, production molecular dynamics, and analysis of results.

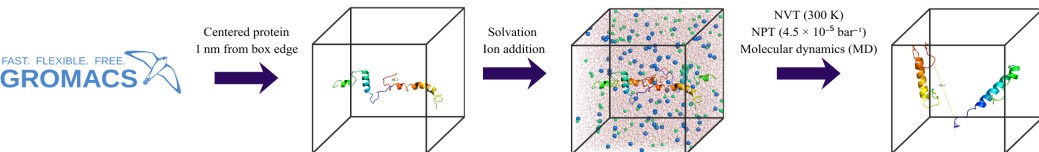

## 3 RESULTS AND DISCUSSION

The Root Mean Square Deviation (RMSD) was calculated considering only the alpha carbon atoms ($C\alpha$) of the $A\beta40$, $A\beta42$, and the $A\beta(40+42)$ system in water, within the interval of 0 to 100 ns, as shown in Figure 3. For the $A\beta40$ peptide, the RMSD showed initial stabilization between $12.0$ and $22.0$ ns, with values around $1.0$ nm. Throughout the simulation, the system exhibited moderate fluctuations and a new stabilization between $40.0$ and $75.0$ ns. The average RMSD value for this system was $0.900$ nm $\pm 0.159$ nm. After this period, fluctuations occur, and the system stabilizes again between $40.0$ and $75.0$ ns, also around $1.0$ nm. In contrast, the $A\beta42$ peptide increased at the beginning of the trajectory, reaching a steady state after approximately $30.0$ ns. From this point on, the values remained between $0.9$ and $1.1$ nm until the end of the simulation. The average RMSD obtained for $A\beta42$ was $0.916$ nm $\pm 0.163$ nm.

Figure 3: Root Mean Square Deviation (RMSD) of the three simulations over $100$ ns. $A\beta40$ (purple line, $0.900$ nm $\pm 0.159$ nm); $A\beta42$ (blue line, $0.916$ nm $\pm 0.163$ nm); $A\beta40 + A\beta42$ (green line, $2.524$ nm $\pm 1.021$ nm).

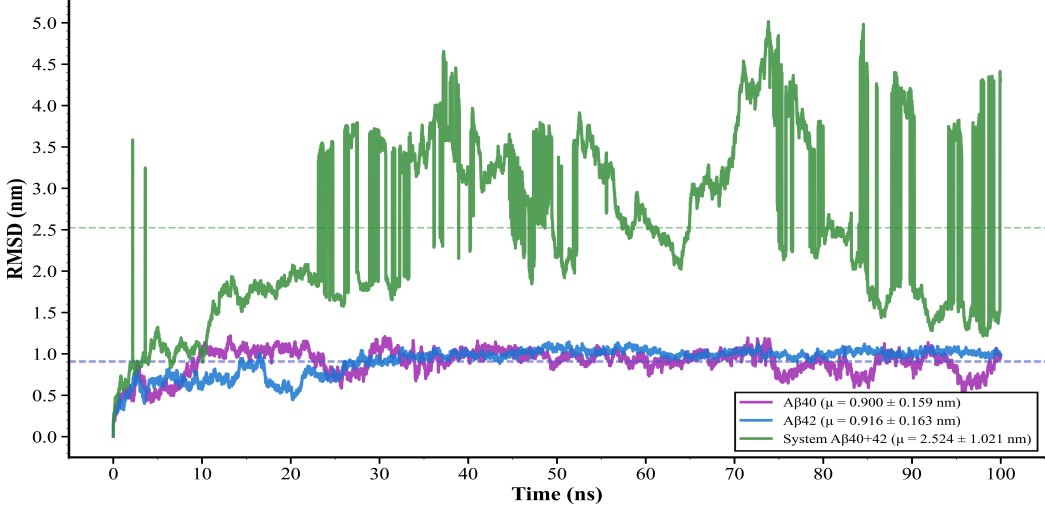

In the case of the $A\beta(40+42)$ system, the RMSD values are higher and more oscillatory, ranging from $2.0$ nm to $4.3$ nm. Stability is observed in the interval from $24.0$ to $35.0$ ns, around $3.8$ nm, with fluctuations from $34.0$ to $51.0$ ns. After $52.0$ ns, the system presents instability and stabilizes again from $75.0$ ns onwards, with values exceeding $4.0$ nm. The average RMSD for this system was $2.524$ nm $\pm 1.021$ nm.

The structural evolution of $A\beta40$ throughout the simulation, presented in Figures 3 and 4, is consistent with the behavior observed in RMSD. The average RMSD value indicates that, despite the peptide's flexibility, no significant global conformational changes occur during the simulation. Secondary structure analysis of the initial $A\beta40$ conformation revealed a predominance of the $\alpha$-helix structure, which corresponds to approximately $40\%$, and the absence of $\beta$-sheet structures. At the end of the simulation, an increase in helical content was observed to $46.42\%$, while the $\beta$-sheet structure remained non-existent ($0.002\%$), as shown in Figure 4. These results suggest that the conformational variations observed over time are associated with the stabilization of the helical structure, without significant formation of $\beta$-sheets (Hoang Viet & Suan Li, 2012; Yang et al., 2009).

For $A\beta42$, the structural evolution throughout the simulation, presented in Figures 3 and 4, is also consistent with the behavior observed in RMSD. The initial increase in RMSD indicates a conformational rearrangement in the early stages of the trajectory, associated with the peptide's adaptation to the simulation environment. The stabilization of the RMSD demonstrates that the total time of $100$ ns was sufficient for conformational equilibrium to occur in aqueous solution (Yang et al., 2009; Reva et al., 1998). The secondary structure analysis of the initial conformation indicated a $\alpha$-helix structure of around $40.5\%$ and the absence of $\beta$-sheets. At the end of the simulation, the $\alpha$-helix structure showed a reduction to $30.44\%$, while the emergence of $\beta$-sheet structures occurred, at approximately $7.44\%$, as observed in Figure 4. This behavior indicates local conformational rearrangements in regions of greater peptide flexibility.

Figure 4: Structures of the $A\beta40$, $A\beta42$ and $A\beta(40+42)$ peptide throughout the molecular dynamics (MD) simulation. Times of 0 ns, 50 ns, and 100 ns.

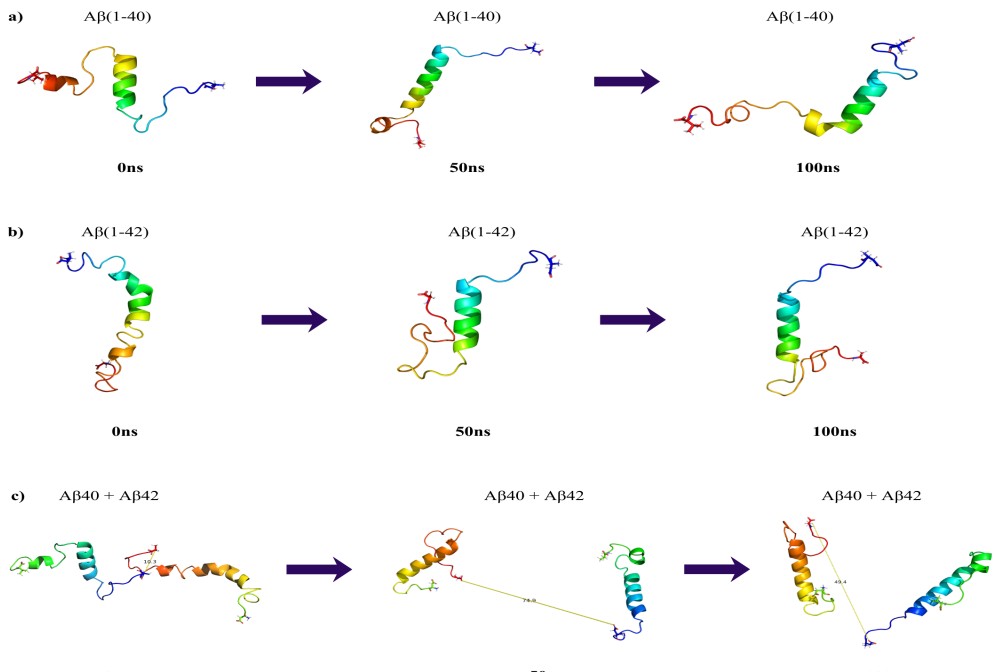

The Root Mean Square Fluctuation (RMSF) values, calculated considering only the alpha carbon atoms ($C_\alpha$), showed differences along the peptide chain. For $A\beta40$, the RMSF values ranged between $0.40$ and $1.40$ nm, with residues ASP1 and HIS13 to LYS16 presenting $RMSF > 0.40$ nm, while residues ASP7 to GLU11 varied between $0.40$ and $0.53$ nm. In the case of $A\beta42$, the RMSF values ranged between $0.33$ and $1.03$ nm, with residues ASP1 and HIS13 to LYS16 exhibiting $RMSF > 0.35$ nm, and residues ASP7 to GLU11 showing values between $0.41$ and $0.65$ nm. For the $A\beta40 + A\beta42$ system, the RMSF values were higher, ranging from $1.40$ to $2.67$ nm across the structures (Turner & et al., 2019).

The RMSF profiles indicate greater flexibility in residues ASP1 and HIS13–LYS16. $A\beta42$ presents lower fluctuations compared to $A\beta40$, suggesting greater structural compactness. In contrast, the

$A\beta40 + A\beta42$ complex exhibits higher RMSF values along both chains, associated with greater conformational instability related to intermolecular interactions.

The Radius of Gyration ($R_g$) values presented a behavior similar to that observed for the RMSD throughout the simulations. For $A\beta40$, the $R_g$ varied between 1.1 and 2 nm, with stabilization observed after approximately 30 ns. Throughout the simulation, occasional fluctuations were recorded, with transient peaks exceeding 2.0 nm. The average $R_g$ value for this system was 1.557 nm $\pm$ 0.220 nm. In $A\beta42$, the $R_g$ showed values between 2.0 and 1.3 nm, also stabilizing after about 30 ns. This system exhibited the lowest average $R_g$ value among those analyzed, equal to 1.447 nm $\pm$ 0.202 nm – Figure 6.

For the $A\beta40 + A\beta42$ system, the $R_g$ fluctuated over a wider range, varying between 1.6 and 6.0 nm along the trajectory. The average $R_g$ value for this system was 3.528 nm $\pm$ 1.292 nm, higher than the values observed for the individual systems.

Figure 5: Root Mean Square Fluctuation (RMSF) of the three simulations over 100 ns. $A\beta40$ (purple line, 0.653 nm $\pm$ 0.232 nm); $A\beta42$ (blue line, 0.916 nm $\pm$ 0.163 nm); $A\beta40 + A\beta42$ (green line, 2.524 nm $\pm$ 1.021 nm). RMSF of the three systems over 100 ns.

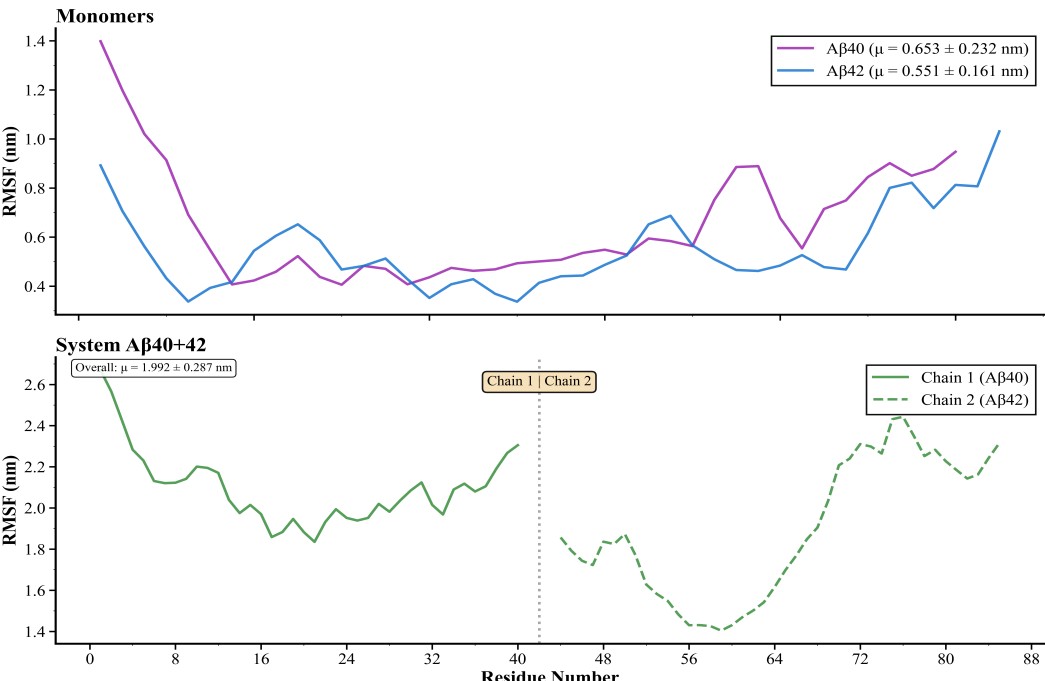

The behavior of the radius of gyration ($R_g$) throughout the simulations provides insights into the structural compactness of the analyzed systems. For the $A\beta40$ and $A\beta42$ peptides, the $R_g$ values remained within a narrow range after the initial accommodation period, indicating the maintenance of an average structural size along the trajectory. $A\beta42$ presented the lowest average $R_g$ value, suggesting a globally more compact conformation compared to $A\beta40$, a result observed in Figure 6 (blue line) and consistent with the smaller fluctuations observed among the simulations (Turner & et al., 2019).

In contrast, the mixed system $A\beta40 + A\beta42$ exhibited higher $R_g$ values and a larger standard deviation. This greater dispersion indicates more evident structural variations throughout the simulation, reflecting changes in the spatial arrangement between the two peptides. The range of $R_g$ variation indicates the occurrence of intermolecular rearrangements, resulting in more extended structures for $A\beta40$ and more compact ones for $A\beta42$ (Figure 6).

The solvent accessible surface area (SASA) represents a crucial parameter for understanding the behavior of peptides in an aqueous environment. The exposure of hydrophobic residues to the solvent constitutes a fundamental thermodynamic driving force for molecular self-association processes, since aggregation allows minimizing unfavorable contact between hydrophobic groups and water

The total SASA reflects the dynamic equilibrium between expanded and compact conformations, with direct implications for the aggregation propensity and amyloid fibril formation (Nguyen & et al., 2021).

The $A\beta40$ peptide exhibited an average SASA of $44.03 \pm 2.05$ nm$^2$, with values ranging from 36.74 to 49.81 nm$^2$. Temporal analysis revealed a pattern of progressive expansion (Figure 7, purple line): $41.90 \pm 1.48$ nm$^2$ in the first 25 ns, increasing to $44.95 \pm 1.41$ nm$^2$ between 25 and 50 ns, stabilizing at $44.00 \pm 1.82$ nm$^2$ between 50 and 75 ns, and reaching $45.26 \pm 1.54$ nm$^2$ in the final period. This increase of approximately 8% in SASA indicates that monomeric A$\beta$40 does not spontaneously converge to compacted structures on a 100 ns time scale, a behavior consistent with experimental data demonstrating a substantially longer lag phase for fibrillization compared to $A\beta42$ (Bitan & et al., 2003).

Figure 6: Radius of Gyration ($R_g$) of the three simulations over 100 ns. $A\beta40$ (purple line, 1.557 nm $\pm$ 0.220 nm); $A\beta42$ (blue line, 1.447 nm $\pm$ 0.202 nm), with the lowest radius of gyration; $A\beta40 + A\beta42$ (green line, 3.528 nm $\pm$ 1.292 nm), with the largest standard deviation observed.

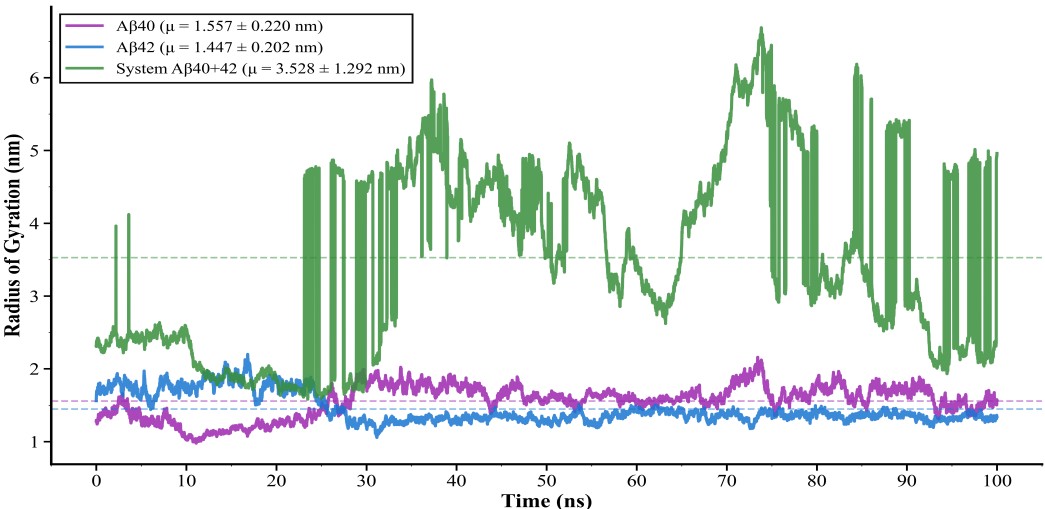

The monomeric $A\beta42$ peptide exhibited an average SASA of $43.66 \pm 2.21$ nm$^2$, a value only 0.8% lower than that of $A\beta40$, ranging from 39.00 to 52.41 nm$^2$. However, the temporal evolution showed the opposite behavior (Figure 7, blue line). The peptide started with expanded structures ($46.65 \pm 1.88$ nm$^2$ in the first 25 ns) and displayed progressive compaction, reaching $42.40 \pm 1.12$ nm$^2$ in the final period. This reduction of approximately 9.1% in SASA indicates a tendency for $A\beta42$ to explore progressively more compact conformations. The difference in temporal patterns between $A\beta40$ and $A\beta42$, despite similar average SASA values, suggests that conformational dynamics are more relevant for aggregation propensity than static average values. Previous studies have demonstrated that $A\beta42$ forms a stable C-terminal $\beta$-hairpin involving residues 31 - 42, a structure absent in $A\beta40$, where the C-terminus is predominantly disordered (Sgourakis et al., 2007).

The complex $A\beta40 + 42$ system presented an average SASA of $84.50 \pm 3.07$ nm$^2$, varying between 73.34 and 93.05 nm$^2$ (Figure 7, green curve). This value corresponds to 96.4% of the sum of individual SASA values ($44.03 + 43.66 = 87.69$ nm$^2$), indicating that only a small fraction of the surface becomes buried at the interface between the peptides. Temporal analysis showed relative stability: $84.18 \pm 2.89$ nm$^2$ at the beginning, $84.50 \pm 2.89$ nm$^2$ between 25 and 50 ns, a slight increase to $86.56 \pm 2.27$ nm$^2$ between 50 and 75 ns, and a reduction to $82.76 \pm 2.89$ nm$^2$ in the final period. The difference of only 3.6% between the complex SASA and the sum of the monomers indicates the formation of a modest interface. This maintenance of relatively independent structures is consistent with the proposed mechanism where $A\beta40$ sequesters $A\beta42$ into dynamic heterotetramers that prevent the formation of larger toxic oligomers (Murray & et al., 2009).

Hydrogen bonds were quantified in three categories: mainchain-mainchain, sidechain-sidechain, and protein-water. The monomeric $A\beta40$ peptide exhibited $14.36 \pm 1.97$ mainchain bonds, oscillating

between 5 and 21 (Figure 8A, purple line). Sidechain bonds were scarce, with an average of $1.22 \pm 1.14$, ranging from 0 to 7 (Figure 8B, purple line). Protein-water bonds were substantially more abundant, totaling $121.25 \pm 6.07$ bonds, with values between 98 and 141 (Figure 8C, purple line). The total number was $136.82 \pm 5.72$ bonds. The percentage distribution showed that protein-water interactions represent $88.6\%$ of the total, mainchain bonds $10.5\%$, and sidechain only $0.9\%$. This predominance of solvent bonds reflects the importance of solvation in stabilizing monomeric states, hindering the formation of aggregation nuclei that require substantial desolvation (Tsemekhman & et al., 2007).

Figure 7: Temporal evolution of SASA for the three systems over 100 ns. A$\beta$40 (purple, $44.03 \pm 2.05$ nm$^2$) showing progressive expansion; A$\beta$42 (blue, $43.66 \pm 2.21$ nm$^2$) with gradual compaction; A$\beta$40+42 complex (green, $84.50 \pm 3.07$ nm$^2$) remaining stable. Dashed lines indicate averages.

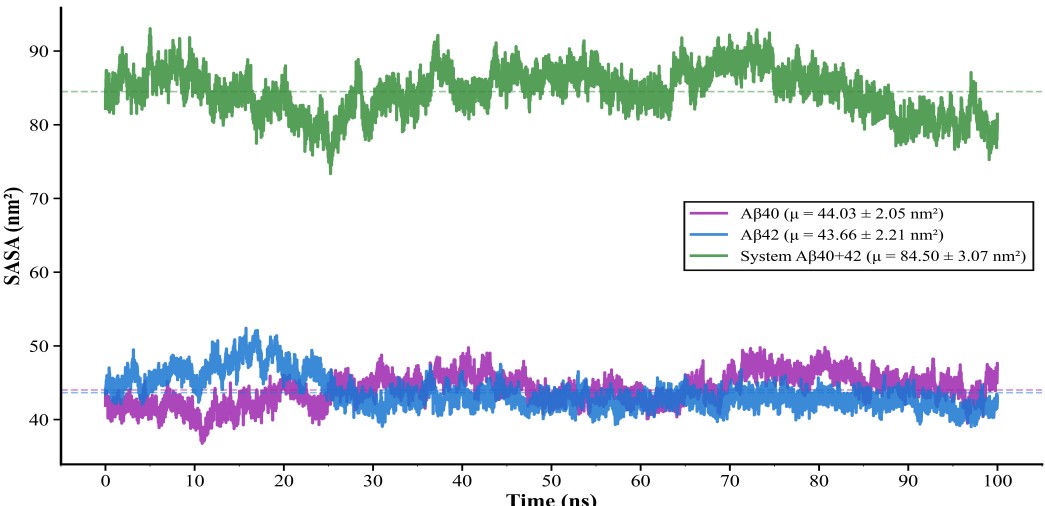

The monomeric $A\beta42 + 42$ peptide presented $13.39 \pm 2.21$ mainchain bonds, ranging from 4 to 21 (Figure 8A, blue line). Temporal analysis showed $12.38 \pm 2.30$ bonds in the first 25 ns, increasing to $14.56 \pm 1.84$ in the final period. Sidechain bonds were even more scarce, with an average of $0.69 \pm 0.92$, ranging from 0 to 5 (Figure 8B, blue line). Protein-water bonds totaled $127.58 \pm 6.26$, ranging from 105 to 152 (Figure 8C, blue line). The total number was $141.66 \pm 5.85$ bonds, a value $3.4\%$ higher than $A\beta40$. The percentage distribution showed an even greater predominance of solvent interactions ($90.1\%$), while mainchain bonds contribute $9.5\%$ and sidechain only $0.5\%$. The finding that $A\beta42$ forms more total bonds than $A\beta40$, despite higher aggregation propensity, is consistent with studies demonstrating that the $A\beta42$ monomer adopts conformations where the hydrophobic C-terminus remains exposed to the solvent, a metastable configuration that facilitates subsequent intermolecular recognition (Yang & Teplow, 2008).

In the complex $A\beta40 + 42$ system, mainchain bonds totaled $28.62 \pm 3.02$, ranging from 16 to 41 (Figure 8A, green line). This value is close to the sum of the individual monomers ($14.36 + 13.39 = 27.75$), suggesting that each peptide retains its intramolecular bond pattern without substantial changes. Sidechain bonds presented an average of $2.65 \pm 1.53$, oscillating between 0 and 9 (Figure 8B, green line). Protein-water bonds were extensive, totaling $247.95 \pm 8.52$, ranging from 208 to 278 (Figure 8C, green line). This value corresponds to the sum of the individual peptides ($121.25 + 127.58 = 248.83$), indicating that complex formation does not significantly alter the solvation pattern. The total number was $279.22 \pm 8.18$ bonds. The percentage distribution maintained a pattern similar to the monomers, with protein-water interactions representing $88.8\%$ of the total.

The maintenance of patterns similar to the sum of the monomers (a difference of only 0.4 total bonds) indicates that the peptides keep their intramolecular interaction networks relatively unchanged in the associated state, without forming cooperative backbone hydrogen bond networks characteristic of amyloidogenic oligomers (Hoang Viet & Suan Li, 2012). This behavior explains

the inhibition mechanism where $A\beta40$ sequesters $A\beta42$ without stabilizing conformations favorable for subsequent aggregation.

Figure 8: Hydrogen bonds in the three systems over 100 ns. (A) Mainchain-mainchain (B) Sidechain-sidechain (C) Protein-water.

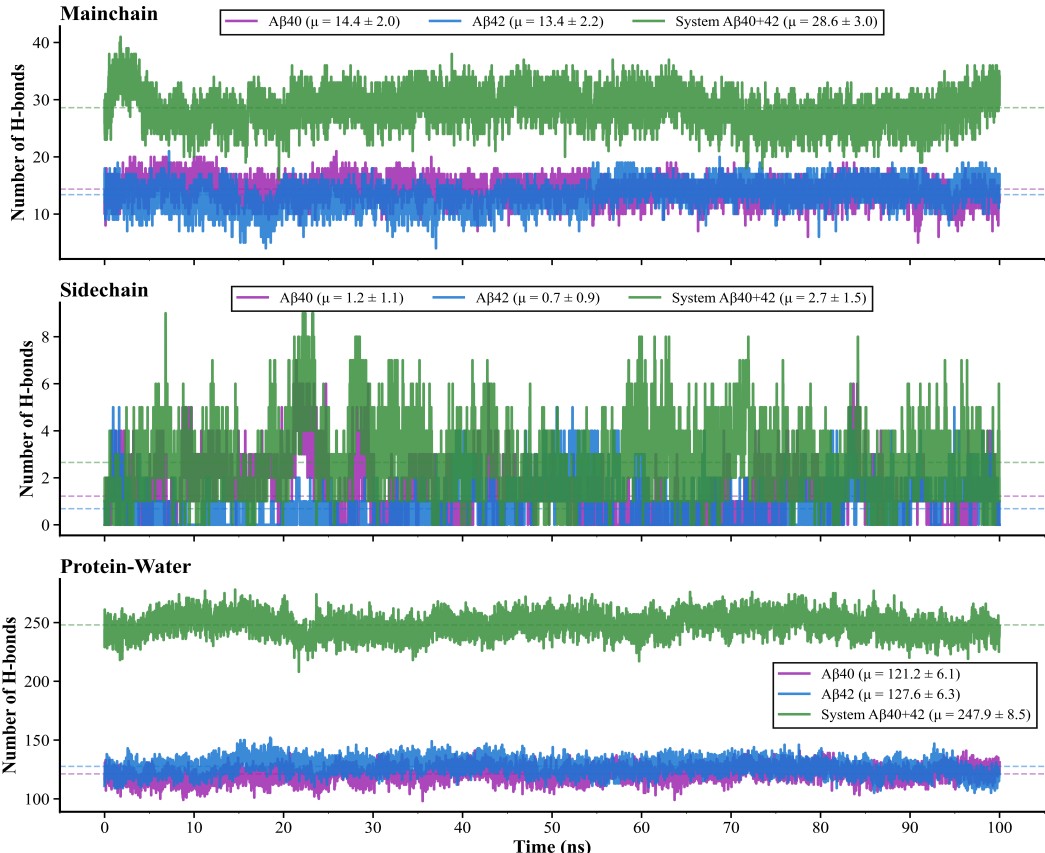

## 4   CONCLUSION

Molecular dynamics simulations demonstrated fundamental differences in the conformational behavior of $A\beta40$ and $A\beta42$ peptides on a 100 nanosecond time scale. Structural characterization revealed that the $A\beta42$ peptide exhibited a propensity for progressive structural compaction and a smaller radius of gyration, consistent with its experimentally observed higher aggregation tendency.In contrast, $A\beta40$ presented structural expansion and greater conformational stability.

The mixed $A\beta40+A\beta42$ system maintained structural characteristics close to the sum of the individual monomers, with a modest intermolecular interface and preservation of intramolecular hydrogen bond networks. These results suggest that the initial co-aggregation between $A\beta40$ and $A\beta42$ does not immediately promote the formation of cooperative structures characteristic of toxic oligomers, providing a molecular basis for the mechanism by which $A\beta40$ can modulate $A\beta42$ aggregation in the early stages of Alzheimer's Disease.

### ACKNOWLEDGMENTS

This study was supported by the Brazilian National Council for Scientific and Technological Development – CNPq (through the Research Productivity Fellowship Level 1A for Claudia do Ó Pessoa [305509/2023-3]); Brazilian Studies and Projects Funding Agency (FINEP)/MCTI/FNDCT (through the Research Project linked to the FINEP - More Innovation Brazil–Health–ICTs–Research, Development and Innovation to Reduce SUS Vulnerabilities and Expand Access to Healthcare program [0361/24]).

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
