# OpenReview forum: "Molecular Dynamics Analysis of $A\beta40$ and $A\beta42$ Peptides and the System $A\beta40$ + $A\beta42$"
_ICLR.cc/2026/Workshop/LMRL — Submitted to ICLR 2026 Workshop LMRL_

### Official Review · Reviewer_nPVe · 2026-02-21
**Sound paper but not the right workshop to host it.**

**Rating:** 3
**Confidence:** 4

**Review:**

The paper proposes an analysis of MD trajectories of two systems and their combination. It tries to quantify how their interaction influences the MD simulations.

**Pros:**
* Their objective is really clear and defined.
* The use of multiple metrics in order to have a complete analysis.
* Good analysis of the behavior, and the conclusion to the impact of those behaviors.

**Cons:**
* Contains zero machine learning, deep learning, or novel representation methodologies, the authors only mention ai in the meaningfulness statement.
*  The works is only an analysis and contains no innovations.
* The initial conditions seem to be arbitraty, and not explained/supported by the authors, making it hard to see the observations and analysis as generalisable.
* 100 ns might be too short for the system to reach convergence, so the conclusion might be too fast.

**Miscellaneous**
* "Throughout the simulation, the system exhibited moderate
fluctuations and a new stabilization between 40.0 and 75.0 ns. The average RMSD value for this
system was 0.900 nm ± 0.159 nm. After this period, fluctuations occur, and the system stabilizes
again between 40.0 and 75.0 ns, also around 1.0 nm." Repetition of the stabilization in this paragraph.

---

### Official Review · Reviewer_udMX · 2026-02-23
**This paper violates double-blind review requirements and must be desk-rejected**

**Rating:** 2
**Confidence:** 4

**Review:**

The paper contains an ACKNOWLEDGMENTS section that explicitly identifies the authors and their institution. Specifically, it names "Claudia do Ó Pessoa" with grant number 305509/2023-3 from CNPq (Brazilian National Council for Scientific and Technological Development) and references a specific FINEP grant 0361/24. A simple search for "Claudia do Ó Pessoa CNPq 305509/2023-3" immediately reveals the author's identity, affiliation, and research group, completely breaking anonymity. This is a clear violation of ICLR's double-blind review policy, which explicitly prohibits any information that could identify the authors during the review process. The acknowledgments section should only be included in the camera-ready version after acceptance, not in the submitted manuscript under review.

---

### Meta-Review · Area_Chair_uEoz · 2026-02-27

**Recommendation:** Reject
**Confidence:** 3

**Metareview:**

Reject.

---

### Decision · Program_Chairs · 2026-03-02

**Decision:**

Reject

**Comment:**

Please see the meta-review.